# Efficacy and Safety of Digital Single-Operator Cholangioscopy in the Diagnosis of Indeterminate Biliary Strictures by Targeted Biopsies: A Systematic Review and Meta-Analysis

**DOI:** 10.3390/diagnostics10090666

**Published:** 2020-09-02

**Authors:** Li-Jia Wen, Jun-Hong Chen, Hong-Ji Xu, Qiong Yu, Kai Liu

**Affiliations:** 1Department of Hepatobiliary and pancreatic Surgery, The First Hospital of Jilin University, Changchun 130021, China; wenlj18@mails.jlu.edu.cn (L.-J.W.); xuhj17@mails.jlu.edu.cn (H.-J.X.); 2College of Clinical Medicine, Jilin University, Changchun 130021, China; chenjh7016@mails.jlu.edu.cn; 3Department of Epidemiology and Biostatistics, School of Public Health, Jilin University, Changchun 130021, China; yuqiong@jlu.edu.cn

**Keywords:** digital single-operator cholangioscopy, SpyGlass™ DS, indeterminate biliary stricture, cholangiocarcinoma, peroral cholangioscopy

## Abstract

Background: Biliary strictures are frequently encountered in clinical practice. The determination of their nature is often difficult. This study aims to systematically evaluate the efficacy and safety of the second generation of digital single-operator cholangioscopy (SpyGlass DS, DSOC) in indeterminate biliary strictures (IBDS) through biopsies. Methods: All relative studies published in Medline, the Cochrane Library, Web of Science, and EMBASE were included. The diagnostic tests for IBDS were compared to the surgical histology, autopsy, or long-term clinical follow-up. The methodological quality of the included studies was evaluated by the Quality Assessment of Studies of Diagnostic Accuracy Included in Systematic Reviews (QUADAS-2). Results: A total of 11 studies, which involved 356 patients diagnosed through biopsies, were included. The pooled sensitivity, specificity, positive likelihood ratio, negative likelihood ratio and diagnostic odds ratio were 0.74 (95% CI: 0.67–0.80), 0.98 (95% CI: 0.95–1.00), 10.52 (95% CI: 5.45–20.32), 0.31 (95% CI: 0.23–0.41), and 65.18 (95% CI: 26.79–158.61), respectively. The area under the curve (AUC) was 0.9479, and the pooled adverse event rate was 7%. The sensitivity and specificity in the heterogeneity analysis were *I*^2^ = 48.1% and *I*^2^ = 25.4%, respectively. Conclusion: SpyGlass DS is a safe and effective technique for IBDS. However, future randomized trials are needed to determine optimal number of biopsies.

## 1. Introduction

Biliary strictures are frequently encountered in clinical practice. Other than iatrogenic biliary strictures, the determination of the etiology of biliary strictures is often difficult. A biliary stricture with an etiology that cannot be ascertained by the appropriate laboratory test, imaging study and endoscopic retrograde cholangiopancreatography (ERCP)-guided tissue biopsy is called an indeterminate biliary stricture (IBDS). Previous studies have revealed that approximately 20% of patients with suspected malignant biliary strictures, who underwent surgical treatment, had benign diseases on the histopathology [1]. Conventional ERCP, along with standard brush cytology and forceps biopsy, is the most commonly used technique for its diagnosis. However, the sensitivity of ERCP for IBDS is <60%, and ERCP plus fluorescence in situ hybridization (FISH) for IBDS is approximately 60%, with a high false negative rate [2]. Endoscopic ultrasound-guided fine needle aspiration (EUS-FNA) is also a useful diagnostic technique. However, this has high accuracy for mainly detecting pancreatic disorders, and only has a sensitivity of 43–86% for IBDS [3].

The fiber-optic single-operator choledochoscope system (Spyglass Optical Direct Visualization System, Boston Science, Marlborough, MA, USA) was first introduced by Chen in 2007 [4]. In 2015, the novel digital single-operator cholangioscope (DSOC), SpyGlass™ Direct Visualization system (SpyGlass DS, Boston Scientific, Marlborough, MA, USA), was introduced. This can provide digital imaging that enables up to a four-fold higher resolution when compared to the previous choledochoscopy system. Furthermore, this has a 60% wider field of view with a tapered tip, which allows for easier scope insertion into the biliary tract [5,6].

In a recent [7] meta-analysis, the accuracy of the DSOC by visual impression (VI) in the diagnosis of IDBS was analyzed. However, to date, the role of DSOC biopsies has not been systematically discussed.

## 2. Methods

The present study was conducted according to PRISMA (the Preferred Reporting Items for Systematic Reviews and Meta-Analyses guidelines) guidelines [8].

### 2.1. Search Method

All studies that reported the utility of DSOC in IBDS, and that were published in the Web of Science (http://www.webofknowledge.com/ 10 April 2020), EMBASE (www.embase.com/ 10 April 2020), MEDLINE (http://isiknowledge.com/medline 10 April 2020), and the Cochrane Library databases (https://www.cochranelibrary.com/ 10 April 2020) from October 2015 (the time when SpyGlass DS was introduced for the first time) to April 2020 were retrieved. The search terms were as follows: (((“Biliary System” OR “common bile duct” OR “Biliary Tract” OR biliary) AND (Neoplasm* OR Cancer OR Malignancy*)) OR cholangiocarcinoma) AND (SpyGlass OR “SpyGlass DS” OR “single-operator cholangioscopy” OR “digital single-operator cholangioscopy” OR “peroral Cholangioscopy” OR DSOC).

### 2.2. Eligibility Criteria

Inclusion criteria: (1) original studies that included patients with IBDS assessed by DSOC, and studies that involved both DSOC and SOC were included when the data of IBDS patients could be separated and extracted; (2) studies written in the English language, with full texts and related data readily available; (3) studies that used a control arm (gold standard), including the histopathology of the surgical specimen, autopsy, or long-term clinical follow-up; and (4) studies that had at least 10 patients.

Exclusion criteria: (1) meeting reports, case reports, letters, reviews and comments; (2) incomplete reports of important data; (3) duplicate publications; or (4) data that involved DSOCs that could not be extracted from previous choledochoscopy systems.

### 2.3. Data Extraction

Two independent researchers screened the literature and extracted the data according to the inclusion and exclusion criteria. In case of differences, a third researcher participated in the discussion and resolution. The extracted contents included the general characteristics of the studies (year of publication, authors, country of origin, type of study, sample size, and follow-up time), patient information (age, gender, etc.), and procedure details (diagnostic yield, indices of accuracy, and adverse events).

### 2.4. Quality Assessment

The quality of the methodology included studies evaluated using the Quality Assessment of Studies of Diagnostic Accuracy Included in Systematic Reviews (QUADAS-2) tool. [9].

### 2.5. Outcomes, Definitions and Gold Standard

IBDS was defined as a biliary stricture with an etiology that could not be diagnosed after all relevant laboratory tests, imaging studies, and ERCP-guided tissue biopsies. The main outcome measurements were as follows: (1) the sensitivity and specificity of the DSOC biopsy, and the raw data of the true positive (TP), false positive (FP), false negative (FN), and true negative (TN) values were extracted; (2) adverse events that included post-ERCP cholangitis, pancreatitis, post-sphincterotomy bleeding, perforation, abdominal pain, fever, nausea and vomiting. Diagnostic gold standard: histopathological findings of the surgical biopsy; if surgical histology was not available, the autopsy or long-term clinical follow-up was used to make a definitive diagnosis.

### 2.6. Statistical Analysis

The Meta-Disc (version1–4) software (Madrid, Spain) was used to evaluate the heterogeneity, which included the threshold effect and non-threshold effect. Evaluation of the threshold effect: The image formed by the accurate estimator of each study in the plane map of the (SROC) curve for the working characteristics of subjects indicated that there was a threshold effect when this was a typical “shoulder-arm” distribution. The non-threshold effect was evaluated by Cochran Q, and the quality was evaluated using the *p*-value and *I*^2^-index. Pooling was performed using the random effects model. If there was statistically significant heterogeneity, meta-regression and subgroup analysis were performed to analyze the source of heterogeneity. The publication bias of each included study was evaluated by Begg’s test and Egger’s test.

## 3. Results

### 3.1. Search Results

A total of 938 related studies were identified from the database. Among these, 462 repetitive literatures were removed and 442 studies were excluded after the title and abstract review for not meeting the selection criteria. Furthermore, 132 articles were excluded, because these did not report the diagnostic accuracy of DSOC by SpyBite in the diagnosis of IDBS. The remaining 54 studies were evaluated, and a total of 11 studies [10,11,12,13,14,15,16,17,18,19,20] that satisfied the selection criteria, which involved 323 patients for SpyBite biopsy (SB), were included in the present meta-analysis (Figure 1).

### 3.2. Characteristics of the Included Studies

Four studies were from the United States [11,12,16,18], five studies were from Europe [10,13,14,15,20], one study was from Asia [17], and one study was from multiple continents [19]. Furthermore, five studies were prospective studies [11,16,17,19,20], two studies were prospective randomized trials [11,19], and six studies were retrospective studies [10,12,13,14,15,18]. In addition, four studies were multicenter trials [15,16,18,19]. The data for the mean age and gender were not available in some studies [10,14,15,16,18], because these also included patients with biliary stones. The basic characteristics are presented in Table 1.

### 3.3. Assessment of the Bias and Quality of Studies

The risk of bias associated with patient selection was low in most of the studies. There was no bias in the applicability of patient selection. Furthermore, there was no bias in the use of the index test in any of the studies. The risk of bias in determining the appropriate reference standard was low for most studies (Figure 2).

#### 3.3.1. Meta-Analysis

Diagnostic accuracy of DSOC for IDBS: The SROC curve analysis did not exhibit a typical “shoulder arm” distribution, indicating that the curve was symmetrical, and that there was no threshold effect. The Spearman’s correlation was 0.173, with *p* = 0.611. The heterogeneity test results for other sources were as follows: sensitivity (SEN): *X*^2^ = 19.28, *p* = 0.037, *I*^2^ = 48.1%; specificity (SPE): *X*^2^ = 13.41, *p* = 0.202, *I*^2^ = 25.4%; diagnostic odds ratio (DOR) Cochran-Q = 5.08, *p* = 0.886, *I*^2^ = 0.0%. This shows that statistical heterogeneity should not be ignored. The pooled sensitivity was 0.74 (95% CI: 0.67–0.80), the pooled specificity was 0.98 (95% CI: 0.95–1.00), the pooled positive likelihood ratio (PLR) was 10.52 (95% CI: 5.45–20.32), the pooled negative likelihood ratio (NLR) was 0.31 (95% CI: 0.23–0.41), the pooled DOR was 65.18 (95% CI: 26.79–158.61), and the area under the curve (AUC) was 0.9479. The sequential removal of each study from the analysis did not significantly affect the effect size or overall heterogeneity of any of the results, indicating that the stability of the results is good (Figure 3).

With the advancements in technology, more and more relevant studies have been recently published. Hence, the investigators also performed an updated meta-analysis to determine the role of VI in the etiological diagnosis of IBDS (Appendix A). The pooled sensitivity, specificity, PLR, NLR and DOR were 0.95 (95% CI: 0.91–0.97), 0.92 (95% CI: 0.88–0.95), 9.29 (95% CI: 5.12–16.86), 0.08 (95% CI: 0.05–0.14) and 178.24 (95% CI: 80.10–396.64), respectively, and the area under the curve (AUC) was 0.9774 (Appendix A). These findings are consistent with that of previous published articles [8], highlighting the diagnostic value of VI during DSOC.

Most of the patients are finally diagnosed with cholangiocarcinoma, chronic cholangitis, IgG4-related cholangitis, and primary sclerosing cholangitis.

Adverse event rate: The pooled adverse event rate was 7% (95% CI: 3–12%, *I*^2^ = 81%) (Figure 4). The main complications were post-ERCP pancreatitis, cholangitis, bleeding and perforation. Acute cholangitis was the most common complication (1.8%; 95% CI: 0.8–3.1%; *I*^2^ = 76.4%). No serious complications or mortality were reported.

#### 3.3.2. Meta-Regression and Subgroup Analysis

The heterogeneity caused by non-threshold effect was present in both the VI and SB groups, while the heterogeneity was maximal in the sensitivity for the SB group (*p* = 0.037, *I*^2^ = 48.1%). Overall, among the factors that may affect the heterogeneity (region, age, sample size and research method), the retrospective study design (*p* = 0.037, *I*^2^ = 57.1%) was found to be the main cause of heterogeneity.

#### 3.3.3. Publication Bias

Begg’s test revealed that there was no statistically significant bias (Z = 1.40, Pr > IzI = 0.161). Egger’s test revealed that there may be publication bias (*t* = 2.40, *p* = 0.040 (95% CI: 0.24–8.06, did not contain 0)).

## 4. Discussion

The differentiation between benign and malignant biliary strictures is often clinically difficult, regardless of the use of different diagnostic methods, as summarized in Table 2 [21,22,23,24,25,26]. Navaneethan et al. [22] reported that the sensitivity of ERCP brush cytology and biopsy was 45% and 48.1%, respectively. Despite the use of the combination of these technologies, the sensitivity only increased to 59.4%. A prospective randomized trial compared the diagnostic value of conventional ERCP and DSOC [20]. The reported sensitivity of ERCP by VI and DSOC by VI was 66.7% and 95.5%, respectively (*p* = 0.02), and the specificity was 66.7% and 95.5%, respectively (*p* = 0.91). The reported sensitivity for ERCP brush cytology and DSOC biopsy was 21.4% and 68.2%, respectively (*p* < 0.1), and the specificity was 100% and 100%, respectively (*p* = 0.99).

In the present meta-analysis, the pooled sensitivity, specificity, PLR and NLR of DSOC were 74%, 98%, 10.52 and 0.31 for SB, respectively. DSOC had a higher sensitivity than FSOC of 74% vs. 60.1% for SB [21], indicating that DSOC has a higher diagnostic yield than FSOC. Mizrahi et al. [27] reported that DSOC in IDBS has a significantly higher diagnostic yield (78% vs. 37%, *p* = 0.004), shorter processing time and lesser radiation when compared to FSOC. Dimas et al. [13] reported that DSOC has higher overall technical success and diagnostic yield for malignancy than FSOC. Furthermore, DSOC with VI is highly sensitive, while SB is highly specific. The combined use of both VI and SB can accurately diagnose the property of IDBS and prevent unnecessary surgeries.

During DSOC, it is important to take enough number of biopsies. Six studies had data on the number biopsies [11,14,15,17,18,19], while two studies reported the mean biopsy number to be more than two. In the subgroup analysis, the sensitivity of the group with more than two biopsies [14,18] and less than two biopsies [11,15,18,19] was 0.75 (95% CI: 0.68–0.81) and 0.69 (95% CI: 0.49–0.85), respectively, indicating that sufficient biopsy tissues are necessary for an accurate diagnosis. However, future randomized trials are needed to determine the optimal number of biopsies.

The present meta-analysis revealed that DSOC had a low adverse event rate of 7% (95% CI: 3–12), suggesting that DSOC is a safe technique for the diagnosis of IBDS. Korrapati et al. [28] reported that peroral cholangioscopy has a low adverse event rate of 7% (95% CI: 6–9). However, this included a variety of peroral cholangioscopes not limited to DSOC. Another study [29] revealed that DSOC has a pooled adverse event rate of 5.2% (95% CI: 2.2–9.2%), in which most of the patients had difficult biliary stones. The major complications of DSOC include cholangitis, pancreatitis, post-sphincterotomy bleeding, and perforation.

There is no standardized visual classification system for digital single-operator cholangioscopy. Sethi et al. [30] reported that the interobserver agreement (IOA) among different endoscopic operators is poor, with an accuracy of 45%. Recently, the Monaco classification has been reported to address this problem. In this study [31], the visual standards were merged into eight categories: stricture, lesion, mucosal features, papillary projections, ulceration, abnormal vessels, scarring, and pronounced pit pattern. The IOA and diagnostic accuracy of DSOC significantly improved with this classification when compared to previous studies.

The heterogeneity of these studies was tested in the present meta-analysis according to country, study design, number of biopsies, sample size and previous ERCP. Prospective studies, single-center studies, studies with a sample size of <60, and studies with previous ERCP biopsies had a high heterogeneity of 46.2%, 42.1%, 41.5% and 38.0%, respectively.

According to the results for the publication bias, it was found that although the SB group had publication bias in the diagnosis of IDBS, the reason could be that most researchers were only interested in positive results. Furthermore, the Begg–Mazumdar indicator and Egger’s test had a low sensitivity when less than 20 studies were included [32].

The present systematic review and meta-analysis is the first to comprehensively assess the efficacy and safety of DSOC-guided target biopsies. Previous studies have systematically explored the diagnostic value of VI during DSOC, but not biopsies [8]. The present study discussed the role of both VI and SB. The combination of Spyglass DS VI and SB can be an effective and feasible method for the etiological diagnosis of IBDS. However, there still is a possibility of false negative results.

The present meta-analysis had some limitations. First, the number of included studies was insufficient. Second, few trials have directly compared the diagnostic performance of DSOC with FSOC, or other methods. Third, future randomized trials are required to determine the standardized visual classification system and the optimal number of biopsies.

## 5. Conclusions

SpyGlass DS is a safe technique with a significantly high diagnostic yield for IBDS when compared to the initial fiberoptic SpyGlass and other diagnostic methods. Furthermore, this has low complication rate. Moreover, this can help avoid unnecessary surgeries for patients with benign strictures.

## Figures and Tables

**Figure 1 diagnostics-10-00666-f001:**
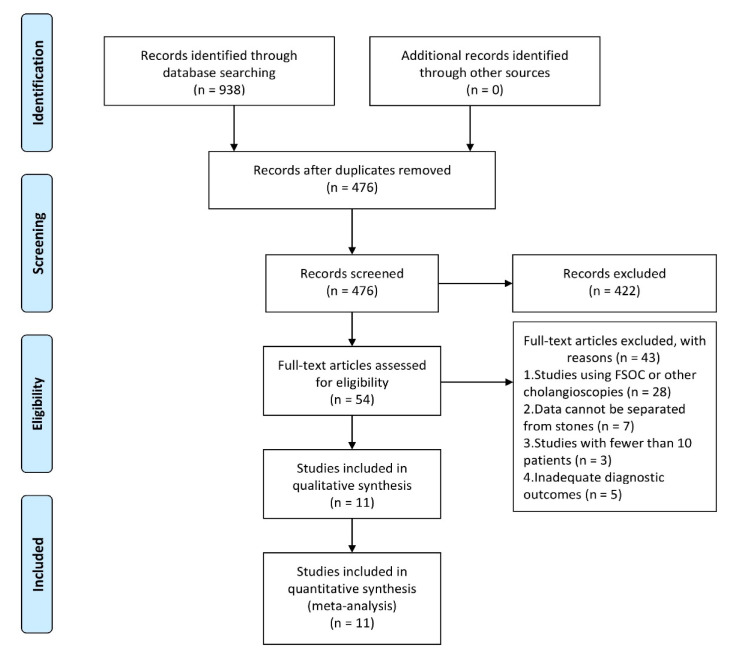
The flow diagram shows the selection process according to the PRISMA statement.

**Figure 2 diagnostics-10-00666-f002:**
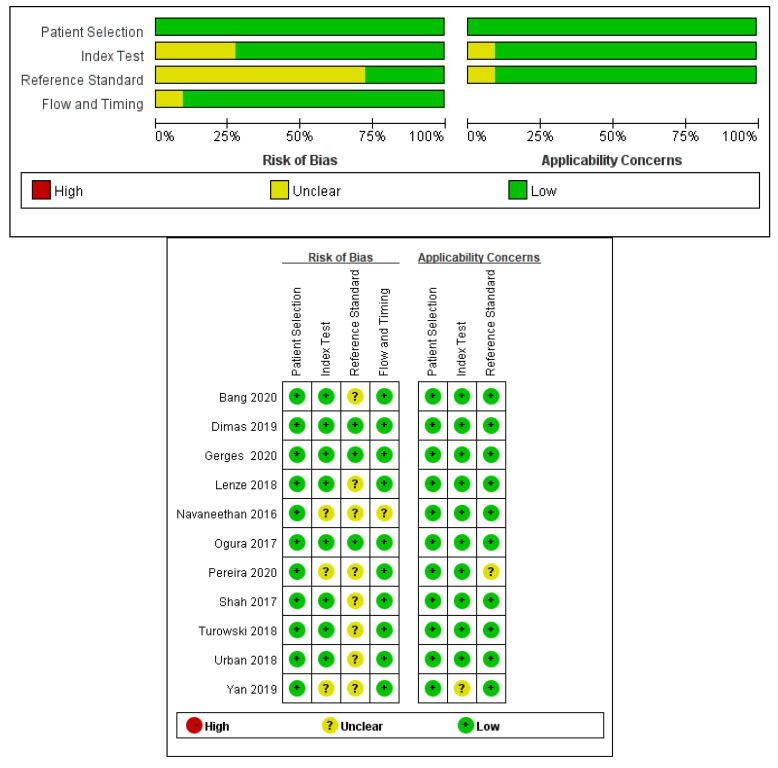
Evaluation of risk of bias and quality of the included studies by the Quality Assessment of Studies of Diagnostic Accuracy Included in Systematic Reviews (QUADAS-2).

**Figure 3 diagnostics-10-00666-f003:**
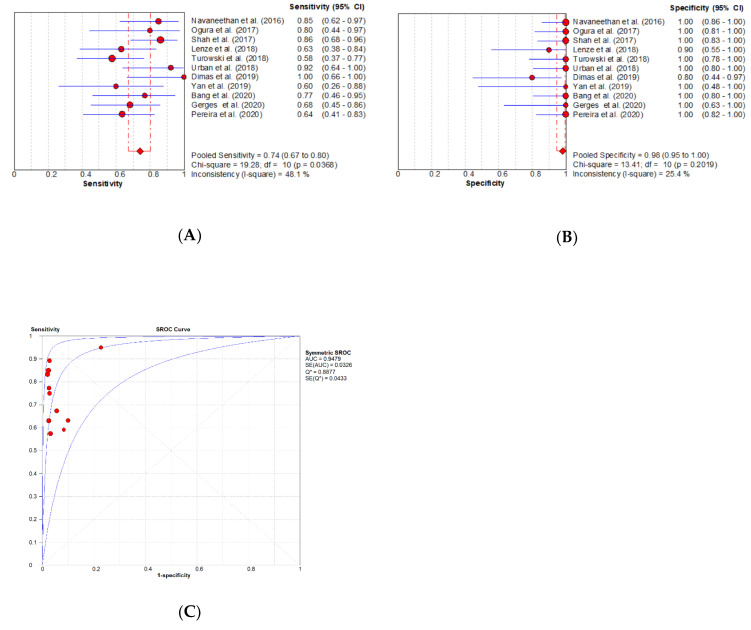
The forest plot of studies that reported the diagnostic yield of digital single-operator cholangioscopy (DSOC) by SB. (**A**) The pooled sensitivity for indeterminate biliary strictures (IBDS) was 0.74 (95% CI: 0.67–0.80); (**B**) the pooled specificity for IBDS was 0.98 (95% CI: 0.95–1.00); (**C**) the summary receiver’s operative characteristics (SROC), with an area under the curve (AUC) of 0.9479.

**Figure 4 diagnostics-10-00666-f004:**
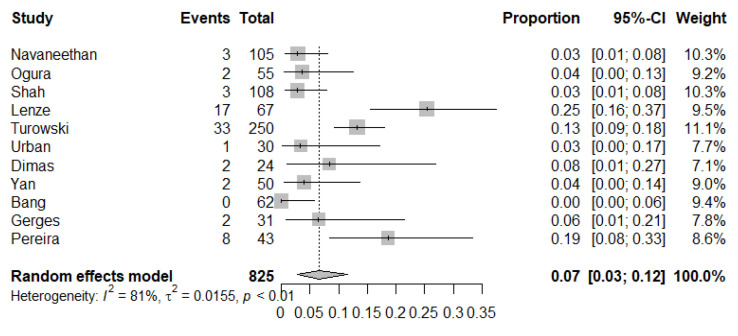
The forest plot of studies that reported a pooled adverse event rate of 7%.

**Table 1 diagnostics-10-00666-t001:** The basic characteristics and diagnostic yield of the included studies for VI and SB.

Study (Year)	Country	Study Design	Age (y) Mean ± SD	Gender, MALE *n* (%)	No. of SB	AEs *n* (%)	No. of Patients		Raw Data of VI				Raw Data of SB			
							VI	SB	TP	FP	FN	TN	TP	FP	FN	TN
Navaneethan et al. [18] (2016)	USA	Retrospective Multicenter	62 ± 15.3 ^‡^	56 (53.3) ^‡^	1 (1–6)	3 (2.9)	44	44	18	1	2	23	17	0	3	24
Ogura et al. [17] (2017)	Japan	Prospective Single center	69 (43–89)	24 (72.7)	2.75	2 (3.7)	30	28	10	2	2	16	8	0	2	18
Shah et al. [16] (2017)	USA	Prospective Multicenter	59.7 ± 8.3 ^‡^	57 (53) ^‡^	NR	3 (2.8)	74	49	28	3	1	42	25	0	4	20
Lenze et al. [14] (2018)	Germany	Retrospective Single center	66 (54–77) ^‡^	38 (56.7) ^‡^	3	17 (25.4)	67	29	24	1	3	39	12	1	7	9
Turowski et al. [15] (2018)	Germany	Retrospective Multicenter	68 ± 14.4 ^‡^	97 (47.1) ^‡^	1.31	33 (13.2)	99	41	42	3	2	52	15	0	11	15
Urban et al. [20] (2018)	Czech	Prospectively Single center	67.3 ± 10.7	19 (63)	5 (0–13)	1 (3.3)	30	30	13	4	0	13	12	0	1	17
Dimas et al. [13] (2019)	Greece	Retrospective Single center	64.2 ± 19.1	15 (62.5)	NR	2 (8.3)	19	19	9	0	0	10	9	2	0	8
Yan et al. [12] (2019)	USA	Retrospective Single center	59.7 (21–86)	15 (68)	NR	2 (4.0)	22	15	10	0	0	12	6	0	4	5
Bang et al. [11] (2020)	USA	RCT Single center	66.4 ± 14.7	38 (61.3)	1 (1–1.5)	0	0	30	0	0	0	0	10	0	3	17
Gerges et al. [19] (2020)	multiple continents	RCT Multicenter	62 (54–79)	17 (54.8)	1.18	2 (6.5)	31	30	21	3	1	6	15	0	7	8
Pereira et al. [10] (2020)	Portugal	Retrospective Single center	61 ± 13 ^‡^	27 (62.8) ^‡^	NA	8 (18.6)	41	41	22	2	0	17	14	0	8	19

‡ Represents data that contain other diseases that cannot be extracted; VI, visual impression; SB, SpyBite biopsy; M, male; AEs, adverse events; No., number; NA, not available; SD, standard deviation; values are mean ± SD (range), median (interquartile range), or *n* (%); TP, true positive; FP, false positive; FN, false negative; TN, true negative; RCT, Randomized Controlled Trial.

**Table 2 diagnostics-10-00666-t002:** The different diagnostic methods used to evaluate the biliary strictures.

Diagnostic Method	Sensitivity (%)	Specificity (%)	Accuracy (%)
DSOC visual impression	95	92	94
DSOC SpyBite biopsy	74	98	85
FSOC visual impression	84.5	82.6	83.5
FSOC SpyBite biopsy	60.1	98	78
DPCS visual impression	83–92	84–92	NA
DPCS biopsy	43–89.5	97	NA
ERCP brush cytology and biopsy	59.4	100	50
ERCP plus FISH	79	91	80
EUS-FNA	43–86	96	NA

DSOC, digital single-operator cholangioscopy; FSOC, fiberoptic single-operator cholangioscopy; ERCP, endoscopic retrograde cholangiopancreatography; FISH, fluorescence in situ hybridization; DPCS, direct peroral cholangiopancreatoscopy system; EUS-FNA, endoscopic ultrasound-guided fine needle aspiration; NA, not available.

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
