# Peer review of "Efficacy and Safety of Digital Single-Operator Cholangioscopy in the Diagnosis of Indeterminate Biliary Strictures by Targeted Biopsies: A Systematic Review and Meta-Analysis"

_diagnostics, 2020, doi:10.3390/diagnostics10090666_

Round 1

Reviewer 1 Report

Thank you for giving me the opportunity for reading your paper. I read with interest your paper. The topic of the paper is interesting and timely for publication. Paper is well written. This is the first paper comprehensively assessing the efficacy and safety of digital single-operator cholangioscopy-guided target biopsies; hence, this systematic review and meta-analysis will be of considerable interest to its readership. Some limitations are clear. Mainly the small number of studies, and this is discussed.

Author Response

Thank you for considering this article for publication in [diagnostics]. We are grateful for the valuable suggestions. Thank you again for your recognition of this article.

Reviewer 2 Report

This is a very interesting study with regard to potential efficacy and safety of Digital Single-Operator cholangioscopy in the diagnosis of indeterminate biliary strictures by targeted biopsies. Therefore, the article should be accepted for publication in its current form under "minor revisions".

  1. In the Introduction section additional data about the analyzed diagnostic modality could be added. The aim of the survey should be more concise.
  2. Grammatical errors should be corrected throughout the Text.
  3. Newly published manuscripts should be included.

Author Response

Thank you for considering this article for publication in [diagnostics]. We are grateful for the valuable suggestions.

Below are the responses to the comments:

Comment 1. In the Introduction section additional data about the analyzed diagnostic modality could be added. The aim of the survey should be more concise.

1.1 We added and modified the data analysis in the Discussion section: and ERCP plus fluorescence in situ hybridization (FISH) for IBDS is approximately 60%, with a high false negative rate [2]. Endoscopic ultrasound-guided fine needle aspiration (EUS-FNA) is also a useful diagnostic technique. However, this has high accuracy for mainly detecting pancreatic disorders, and only has a sensitivity of 43-86% for IBDS [3].

1.2 In the Aim section, we deleted the following content, which was about the first generation cholangioscopy, in order to make the survey more concise: This allowed the choledochoscopy to be performed by a single operator. This system has high diagnostic sensitivity, provides the direct visualization of the biliary and pancreatic ductal system, and has the ability to obtain targeted biopsies. However, the suboptimal image quality is its main limitation.

Comment 2. Grammatical errors should be corrected throughout the Text.

We have re-checked the grammar of the whole text, and corrected the errors.

Comment 3.  Newly published manuscripts should be included.

The present article’s screening time was from October 2015 (the time when SpyGlass DS was introduced for the first time) to April 2020 (when the article was started). We included all articles that met the inclusion criteria during this period.

We would be happy to make any further changes that may be required. Thank you again for the recognition of this article

Reviewer 3 Report

This systematic review is interesting and well described. I think that the discussion about versatility and/or patient’s wishes for DSOCs would add value to this article.

  1. Exclusion criteria. Please clarify the meaning "(4) data that involved DSOCs that could not be extracted from previous choledochoscopy systems.”.

  1. What was the final diagnosis of the patients? If possible, please provide this information.

  1. Are VI diagnostic methods unified among studies? Please explain.

  1. Biliary stones are often accompanied with biliary duct cancer. Moreover, biliary atones may be also important cause of biliary stricture. Why did seven studies need to be excluded from the analysis although authors described “data cannot be separated from stones” in Figure 1? Please clarify.

Author Response

Thank you for considering this article for publication in [diagnostics]. We are grateful for the valuable suggestions.

Below are the responses to the comments:

Comments 1. Exclusion criteria. Please clarify the meaning "(4) data that involved DSOCs that could not be extracted from previous choledochoscopy systems.”

In some articles, the authors conducted an undifferentiated study of patients with first-generation choledochoscopy and SpyGlass DS. However, the data that could not be separated from each other. Furthermore, in the inclusion literature, although two sections of patients were included, each section of data was independent.

 Comments 2. What was the final diagnosis of the patients? If possible, please provide this information.

Most of the patients are finally diagnosed with cholangiocarcinoma, chronic cholangitis, IgG4-related cholangitis, and primary sclerosing cholangitis. We included the final diagnosis of patients at the meta-analysis section.

Comments 3. Are VI diagnostic methods unified among studies? Please explain.

The VI diagnostic methods were not unified among the studies. This was also one of the purposes for writing this article. Furthermore, one of the purposes of this article was to establish and facilitate the establishment of standard visual diagnostic models. At present, some articles, such as The Monaco Classification [29], has been put forward a standard that contains eight observation indexes, and proven to be effective. In the future, more RCTs are needed to verify the most advanced visual diagnostic criteria.

Comments 4. Biliary stones are often accompanied with biliary duct cancer. Moreover, biliary atones may be also important cause of biliary stricture. Why did seven studies need to be excluded from the analysis although authors described “data cannot be separated from stones” in Figure 1? Please clarify.

This review mainly discusses the diagnosis of indeterminate biliary strictures. This is a diagnostic procedure.

In the seven excluded articles, patients with biliary stones were accurately diagnosed with traditional ERCP and defined as refractory stones due to the number, size, and location of the stones. This is a therapeutic procedure, and not what we intended to discuss.

The article “The utility of digital cholangioscopy (SpyGlass DS) in biliary and pancreatic diseases: A clinical feasibility study at two tertiary care centers in Saudi Arabia (with Videos)” [DoI:10.1016/j.ajg.2019.12.003] is an example.

We would be glad to make any further changes that may be required. Thank you again for your recognition of this article.